# Polymorphism and Selection Pressure of SARS-CoV-2 Vaccine and Diagnostic Antigens: Implications for Immune Evasion and Serologic Diagnostic Performance

**DOI:** 10.3390/pathogens9070584

**Published:** 2020-07-17

**Authors:** Eric Dumonteil, Claudia Herrera

**Affiliations:** Department of Tropical Medicine, School of Public Health and Tropical Medicine, and Vector-Borne and Infectious Disease Research Center, Tulane University, New Orleans, LA 70112, USA

**Keywords:** antigenic drift, diversifying selection, coronavirus, immune evasion, diversity

## Abstract

The ongoing SARS-CoV-2 pandemic has triggered multiple efforts for serological tests and vaccine development. Most of these tests and vaccines are based on the Spike glycoprotein (S) or the Nucleocapsid (N) viral protein. Conservation of these antigens among viral strains is critical to ensure optimum diagnostic test performance and broad protective efficacy, respectively. We assessed N and S antigen diversity from 17,853 SARS-CoV-2 genome sequences and evaluated selection pressure. Up to 6–7 incipient phylogenetic clades were identified for both antigens, confirming early variants of the S antigen and identifying new ones. Significant diversifying selection was detected at multiple sites for both antigens. Some sequence variants have already spread in multiple regions, in spite of their low frequency. In conclusion, the N and S antigens of SARS-CoV-2 are well-conserved antigens, but new clades are emerging and may need to be included in future diagnostic and vaccine formulations.

## 1. Introduction

The emergence and rapid spread of a novel Coronavirus, referred to as SARS-CoV-2, is resulting in one of the worst pandemics in the world, causing an unprecedented health and economic crisis. About seven months after the first cases were identified, over 8 million cases have been reported worldwide, with over 400,000 deaths according to the Johns Hopkins Coronavirus Resource Center [1].

The pandemic has triggered multiple efforts at developing serological tests, able to detect both acute infections by detecting virus-specific IgM, as well as recovered individuals by detecting virus-specific IgG. Several immunochromatographic rapid tests are already available [2], and several more are likely to become available by the end of 2020. Such tools will be critical to increase testing for the accurate and rapid identification of cases and their isolation to limit further transmission of the virus. However, their performance needs to be evaluated, and initial testing suggested variable performance of these tests [2,3]. Test performance relies in part on the antigen used, and its conservation among virus strains circulating in the population being tested. Currently, most of these tests are based on the Spike glycoprotein (S) or the Nucleocapsid (N) viral proteins [2]. The receptor-binding domain (RBD) of the S protein, which mediates binding to the angiotensin-converting enzyme 2 (ACE2) receptor in human cells [4], is also widely used as a diagnostic antigen.

Similarly, vaccine development efforts have been very intense and a growing number of vaccine candidates are being quickly moved into clinical trials. These are based on different technological platforms, ranging from recombinant proteins, RNA and DNA vaccines, or recombinant viral vectors [5,6]. A first RNA vaccine candidate recently completed clinical phase 1 evaluation and is expected to move into Phase 2 shortly. Most of these vaccine candidates are based on the viral S protein, or the RBD as antigen. Multiple potential vaccine epitopes have also been identified in the S, as well as in the N viral proteins [7]. As for diagnostics, conservation of these vaccine antigens among viral strains is critical to ensure broad protection and avoid immune evasion by the virus. 

As an RNA virus, SARS-CoV-2 is prone to frequent mutations, in spite of some proof-reading abilities of its RNA polymerase complex [8,9]. An early assessment of genomic changes in SARS-CoV-2 showed a mutation hot-spot in the virus’s RNA-dependent RNA polymerase (RdRp), but a few mutations were also detected in other parts of the viral genome, including the N and S proteins [10]. The growing availability of a large number of complete genome sequences gathered since the beginning of the pandemic provides a unique tool to assess the extent of viral antigen polymorphisms, and potential selection pressures. A first analysis of polymorphisms in the S glycoprotein from the first sequences in December 2019 to 13 April 2020 identified a handful of variant sites, including D614G, S943P, and possibly L5F and L8V [11]. Variant sites V367F, G476S, and V483A were also identified in the RBD. Here, we analyzed the sequence variation in a broader set of viral proteins N and S, which represent the main diagnostic and vaccine antigens to date. We then examined the implications of the identified sequence variants on vaccine and serological diagnostic performance. 

## 2. Results

Analysis of over 17,000 genome sequences confirmed the SARS-CoV-2 is a fast-evolving virus. Indeed, in the less than 5 months that viral sequences have been available, we detected multiple sequence variants, and some virus strains currently circulating in multiple countries have somewhat diverged from the first isolates from December 2019 in Wuhan, China (Figure 1A). Variants were scattered throughout the viral genome, rather than clustered in specific genes (Figure 1B,C), and some sequence variation could be detected within both the N and S genes.

These genes were then analyzed in detail and separately. For the N protein, we included a dataset of 16,656 sequences, and significant sequence diversity was detected, with up to 326 distinct protein sequences. For a clearer assessment of their phylogenetic relationship, these variant sequences were analyzed independently (Figure 2A). Notably, a structuring including up to seven incipient clades was found emerging, with sequences from the first virus from Wuhan, China included in Clade 1 (Figure 2C). There was no specific geographic clustering of the sequence variants, illustrating the widespread multidirectional spreading of the virus across the world. A notable exception was observed for Clade 3, which included mostly sequences from Europe. Analysis of sequence variation along the protein sequence indicated that about half of the protein on the amino side was mostly conserved, except in two regions at sites 13 and 203–204, respectively (Figure 2B). On the other hand, the carboxy half of the protein appeared more variable, but this also reflected some sequencing ambiguities. 

A total of 178/419 (42.5%) sites presented variation in the N protein. This included seven sites with four variant amino acids, seven sites with three variant amino acids, and 13 sites with two variant amino acids that were found under significant diversifying selection pressure (26/419 (6.2%), Table 1). Because of these changes, the N protein appears to be slowly diverging from the sequence from some of the early virus, belonging to Clade 1, and up to six additional major clades (Clades 2–7) are emerging for the N antigen (Figure 2C). Site D144 that can be substituted by E, H, Y, or N may disrupt a predicted epitope (ALNTPKDHI 138–146) [7,12]. Importantly, most variants were still found at relatively low frequency among the viral population (0.018 to 0.541%), with only R203X and P13X variants detected at higher frequency (18.108 and 1.589%, respectively, Table 1), indicating an overall high level of conservation of the N protein. Nonetheless, many of the low frequency variants were found to have already dispersed in multiple countries and regions. For example, S202X variants were detected in 90 cases from Australia, China, Democratic Republic of Congo, England, Ghana, India, The Netherland, Russia, Saudi Arabia, Senegal, Turkey, and the USA. Similarly, D22X was detected in 53 cases from Australia, England, Taiwan, Uruguay, and Wales. On the other hand, a few variants were likely more associated with limited clusters of infection, such as A208G variants, which were mostly limited to the USA so far.

A similar analysis of the S antigen was performed, based on 17,802 sequences. It revealed even greater sequence diversity, with up to 681 unique S protein variants, and most of this diversity was observed in the most recent months of March and April 2020 compared to January and February 2020 (Figure 3A). Furthermore, up to six emerging clades could be defined, that present a clear divergence from Clade 1, which includes some of the first sequences from Wuhan, China (Figure 3B). While some sequences from Clades 3 and 6 could be detected as early as February 2020, sequences from Clades 2, 4, and 5 appeared in March 2020 and expanded in April 2020. At the same time, sequences from Clade 1 appeared less frequent with time (Figure 3A,B).

Analysis of S glycoprotein variation indicated that amino acid variants were spread along most of the protein sequence (Figure 3C), although a few regions of lower sequence conservation could be detected at positions 260–320 just before the RBD, at position 445–515 in the carboxy end of the RBD, and at site 614. Further analysis of each major clade revealed that each had amino acid substitutions that concentrated in different domains of the proteins, except for Clade 1, which accumulated the greatest number of substitutions across the entire protein (Figure 4). For example, Clade 2 had more substitutions between sites 850–970, Clade 3 between 550–750 and 1150–1250, Clade 4 between 250–320, Clade 5 between 140–250 and 420–500, and Clade 6 between 750–800.

A total of 362/1273 (28.4%) sites presented variation in the S protein, of which 30 sites (2.4%) were found under significant diversifying selection pressure (Table 2). These included one site with five variant amino acids, one site with four variants, seven sites with three variants, 11 sites with two variants, and 12 sites with a single variant amino acid. Furthermore, different selection patterns were identified in each major clade, and only a few notable sites had substitutions in more than one clade (Table 2 and Figure 4). For example, sequences from Clade 1 are clearly defined by site D614, which is under strong diversifying selection, together with sites V615, G476, V483, and H519 and their corresponding variants. Most sequences from Clades 2–6 have a D614G substitution, together with clade specific variants. Thus, Clade 2 is characterized by a cluster of substitutions around sites 936–943, with specific sites D936, S940, and S943 and their variants under strong diversifying selection. Clade 3 is characterized by variants sites V622, A653, A684, A703, and their variants, and Clade 5 by sites D215, S221, and Q238 and their variants (Table 2 and Figure 4). A single predicted epitope may be affected by diversifying selection and substitutions at site 1078. The furin cleaveage region (671–692), and particularly the cleavage site, were well conserved, although two sites, Q675 and A684 are under diversifying selection. Substitutions at these less conserved sites may thus not affect furin cleavage, which is unique to SARS-Cov-2 [4]. Similarly, none of the sites with N-linked glycosylation [13] were found under diversifying selection, allowing for the conservation of the glycosylation pattern of the S protein across its diversity. 

With the exception of the D614G substitution which has taken over and is now widespread in virus populations across the globe (over 63% of sequences carry this substitution), the other variants under selection still represent a low proportion of viral sequences, ranging from 0.017 to 0.586% (Table 2). A few of these variants likely correspond to limited clusters of infections, as they come from a single geographic region and are grouped in time. This is the case for the G1124V variant, which is limited to 50 cases from Victoria, Australia, between 20–27 March 2020. Similarly, the N439K variant is limited to 40 cases from Scotland, identified between 16 March–5 April 2020. However, most of the other variants have already spread to multiple countries and regions, such as Q675X, which has been found in Denmark, England, Finland, Iceland, Norway, Scotland, Spain, and the USA over March and April 2020. Similarly, L5F variants have been found on 102 cases from Australia, Belgium, Canada, England, France, Iceland, India, Italy, Japan, Netherlands, Portugal, Scotland, Singapore, Taiwan, Thailand, USA, and Wales, and H49X variants have been found in 36 cases from Australia, China, England, Mexico, Taiwan, and the USA, for example.

As mentioned above, some of the sequence variation affecting the S protein was detected within the RBD, which is a key functional domain of the protein and one of the most used targets for serological diagnostics. We thus analyzed in detail its polymorphism. Sequence analysis of RBD revealed that it represented a highly conserved region of the S protein. Nonetheless, up to 54 RBD sequence variants were identified, with again some significant divergence from the first sequences from Wuhan, China (Figure 5). Importantly, divergence seemed to increase with time as more variants accumulate and become established. A total of seven sites from the RBD were found under significant diversifying selection pressure, and variants sites within the RBD were observed in each of the major clades of the S protein (Table 2). Nonetheless, while possible RBD clades are emerging, these do not match the S protein major clades described above.

## 3. Discussion

Antigen polymorphism from pathogens has the potential to impair serological diagnostic test performance, as well as vaccine efficacy. It is thus of key importance to consider these aspects for serological test and vaccine development, to ensure their usefulness and broad efficacy. This is commonly done for influenza vaccines for example, which are updated each year based on circulating viral strains as cross protection among strains is still elusive [14]. Here, we investigated the sequence diversity of two major antigens of the novel SARS-CoV-2 virus, the N and S proteins. Importantly, a significant level of sequence diversity was detected for both antigens, with incipient clades emerging as multiple sites were found under significant diversifying selection pressure. 

The N protein, mostly used in serological diagnostic tests [2] had a large number of sequence variants, and 6.2% of its residues were found under diversifying selection. Overall, up to seven major sequence clades have been emerging in recent months for this antigen, and these did not show any geographic clustering. A notable exception was Clade 3 of the N protein, which appeared over-represented in sequences from Europe. Importantly, predicted epitopes appeared conserved so far, although a more detailed epitope mapping is still needed for this antigen. Nonetheless, N protein variants diverging from the initial sequences from Wuhan, China are now circulating in most geographic regions. While these changes are so far limited to a relatively small proportion of sequences (23.4%) and may not interfere with protein antigenicity, the inclusion of some of the variants in serological tests would ensure optimum sensitivity of tests, particularly if some of these variants become more frequent. 

The S glycoprotein is the main vaccine candidate currently tested in multiple vaccine platforms/formulation [5,6]. Compared to the N antigen, it is more conserved and only 2.4% of its sites were found under diversifying selection pressure. We confirmed the importance of most of the variant sites previously identified in this antigen. These include D614G and S943P, as well as L5F and L8V and variant sites V367F, G476S, and V483A in the RBD [11]. However, multiple additional variants were also identified here, leading to the identification of up to six major clades of the S glycoprotein that are emerging. Most of these variants appeared in the months of March and April 2020 and may be slowly replacing the virus presenting sequences similar to that of the initial isolates from Wuhan, China. Indeed, while most of the variants still have a low frequency in the viral population, several have already spread to multiple countries and regions, where they may reach higher frequencies in the near future if they are successfully transmitted. Importantly, none of the substitutions identified affected the glycosylation pattern of the S protein, and none of the predicted epitopes appear affected. While the functional impact of these variants is unknown, the D614G mutation has been associated with potential increased viral transmission and/or fitness [11], which may explain why it became so frequent. A recent comparison of functional properties of the S proteins with aspartic acid (SD614) and glycine (SG614) confirmed a greater infectivity correlated with less S1 shedding and greater incorporation of the S protein into the pseudovirion with the SG614 variant [15]. Similar functional studies of the additional variants identified here may help evaluate their impact on virus fitness. Future studies will potentially provide data on how the different clades identified here may be successfully transmitted or go extinct.

While the RBD is particularly well conserved, some sequence variation was also detected in this region within the S glycoprotein, with up to 54 sequence variants. Because these differ by only 1–2 amino acids, the overall antibody recognition of the RBD can be expected to be mostly preserved, but some specific epitopes may nonetheless be lost. In addition, our phylogenetic analysis suggested that possible clades may be emerging within the RBD, as well, and newer sequences may diverge further from the sequence from the initial isolates from Wuhan.

In conclusion, we found that the N and S antigens of SARS-CoV-2 are so far highly conserved, so that both are good antigens for both diagnostic and vaccine development. However, some sequence variation is also emerging and 6–7 phylogenetic clades could be identified for both antigens. Some of these sequence variants have already spread in multiple countries and regions, in spite of their low frequency. Sequence variants may arise by random substitutions in the viral genome during replication, but the significant diversifying selection detected at multiple sites in both antigens suggests that immune selection pressure and adaptation to human hosts may be driving some of these changes [16,17], which may lead to the establishment of some of these variants. New variants are also likely to emerge with time. The recent identification of potential co-infections with more than one viral strain suggests that recombination could also contribute to the generation of SARS-CoV-2 genetic diversity [18]. Therefore, further monitoring of antigen drift over time will be needed to ensure that diverging antigens can be identified in a timely manner and included in future diagnostic and vaccine formulations.

## 4. Materials and Methods

### 4.1. Viral Sequence Data

Whole genome sequences from 18,247 SARS-CoV-2 virus were obtained from GISAID (Appendix A), covering virus isolates from multiple continents, including Asia, Africa, Europe, Oceania, and America. These sequences included those from initial human cases in Wuhan, China, from December 2019 up to sequences from 11 May 2020.

### 4.2. Sequence Analysis

Viral genome sequences were aligned using MAFFT [19] as implemented in Geneious 11, and alignments were edited to exclude partial or low-quality sequences. A final alignment including 17,853 quality sequences were used to construct phylogenetic trees using FastTree [20] for a global analysis of viral diversity across the world. FastTree infers approximately-maximum-likelihood phylogenetic trees. Sequence conservation across genome alignment was calculated using a sliding window analysis in Geneious.

Separate analyses were then performed using S and N genes, as well as the RBD from the S protein (positions 319–540 within the S protein). For these, translated sequences were aligned with the MAFFT algorithm using Blossum62 matrix and the frequency of variants at each site was calculated. Unique sequences from these proteins were then selected, and phylogenetic trees were constructed using FastTree as above. Major clades were defined according to tree structure and branch support measures. Predicted epitopes from these antigens [7,12] were mapped in the alignments, as well as glycosylation sites [13], to assess their conservation among viral sequences. Finally, evolutionary selection pressures on the antigens were analyzed using the Fast, Unconstrained Bayesian AppRoximation (FUBAR), as implemented in HyPhy [21], and statistical significance was considered at a threshold of *p* < 0.1. 

## Figures and Tables

**Figure 1 pathogens-09-00584-f001:**
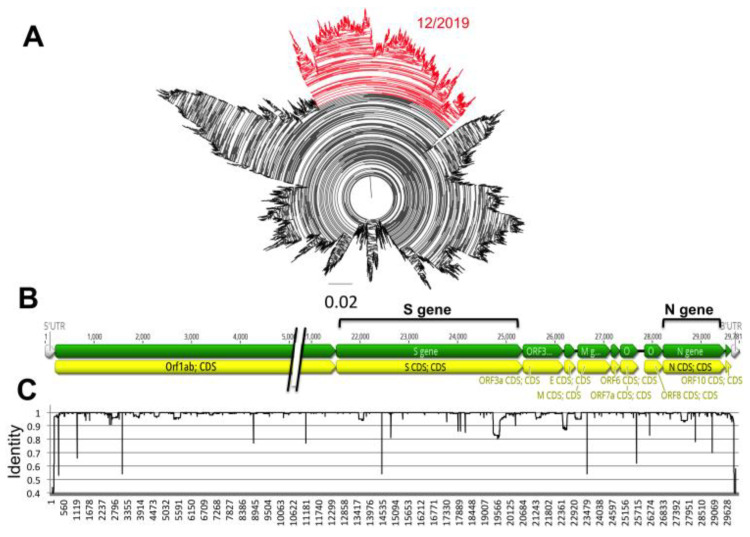
Diversity of SARS-CoV-2 genome. (**A**) Phylogenetic analysis of 17,853 SARS-CoV-2 genomes. Highlighted in red is the clade that includes the earliest sequences derived from human cases in December 2019. (**B**) Diagram of SARS-CoV-2 genome organization. Position of the Spike glycoprotein (S) and Nucleocapsid (N) genes is highlighted. (**C**) Sliding window analysis of nucleotide identity along SARS-CoV-2 genome.

**Figure 2 pathogens-09-00584-f002:**
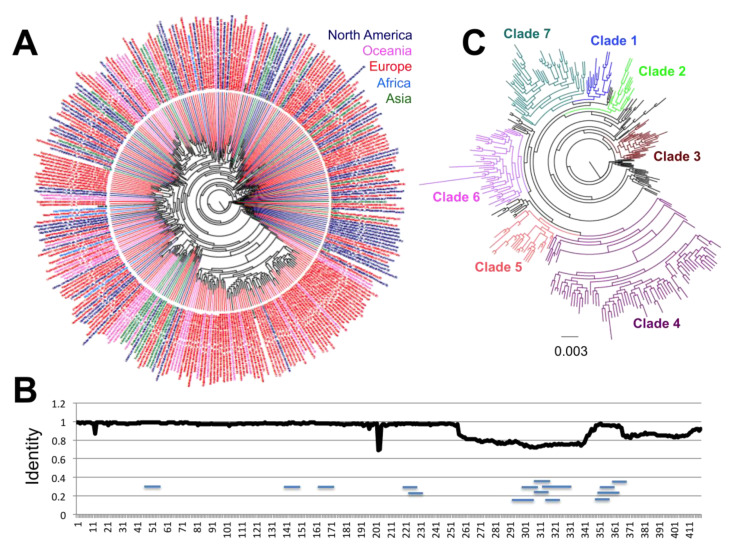
Sequence diversity of SARS-CoV-2 N antigen. (**A**) Phylogenetic analysis of 326 unique N protein sequence variants, color-coded according to region of origin. (**B**) Sliding window analysis of sequence identity along the N protein sequence. Small horizontal lines within the sequence indicate the position of predicted epitopes. (**C**) Phylogenetic analysis showing the identified incipient clades. Early sequences from Wuhan, China, from December 2019 are included in Clade 1.

**Figure 3 pathogens-09-00584-f003:**
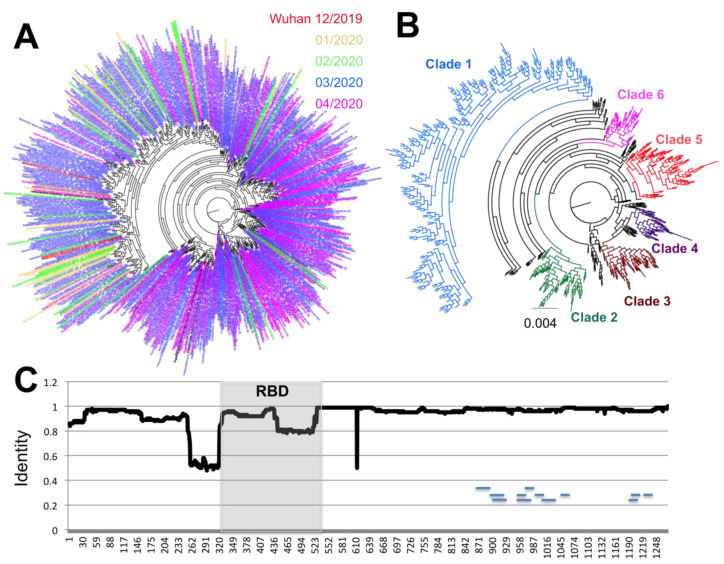
Sequence diversity of SARS-CoV-2 S antigen. (**A**) Phylogenetic analysis of 681 unique S antigen sequence variants, color-coded according to date of identification. (**B**) Phylogenetic analysis showing the identified incipient clades. Early sequences from Wuhan, China, from December 2019 are included in Clade 1. (**C**) Sliding window analysis of sequence identity along the S protein sequence. Small horizontal lines within the sequence indicate the position of predicted epitopes. The receptor-binding domain (RBD) is highlighted in gray.

**Figure 4 pathogens-09-00584-f004:**
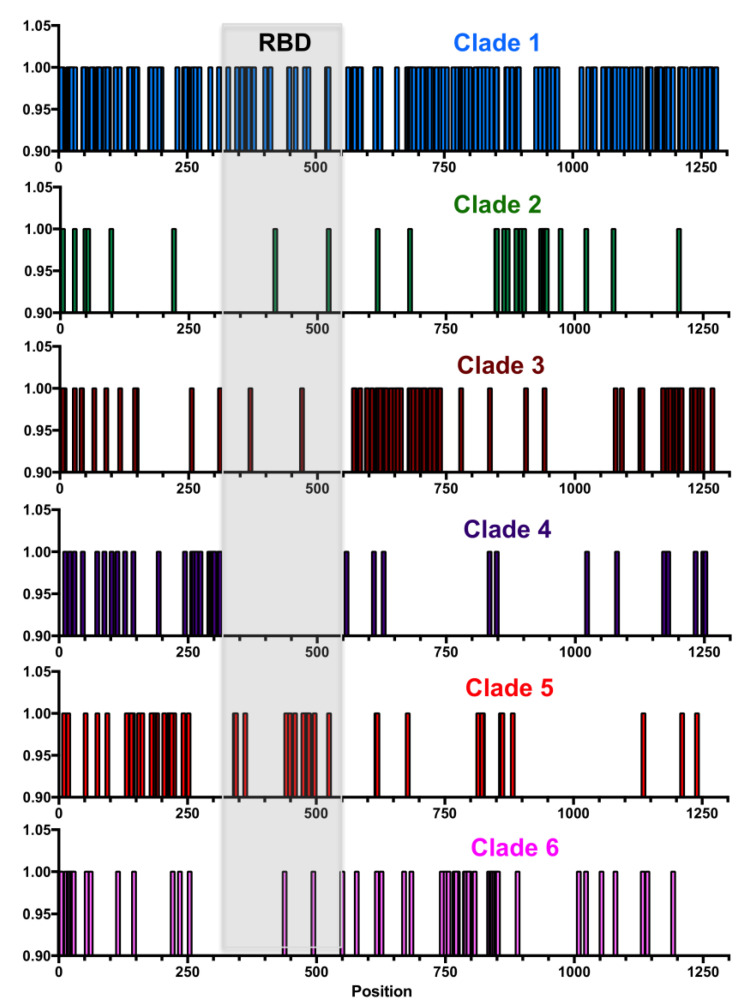
Location of variant sites along the S antigen sequence for each clade. Each vertical bar indicates a variant site. The position of the RBD within the S antigen is highlighted in light gray.

**Figure 5 pathogens-09-00584-f005:**
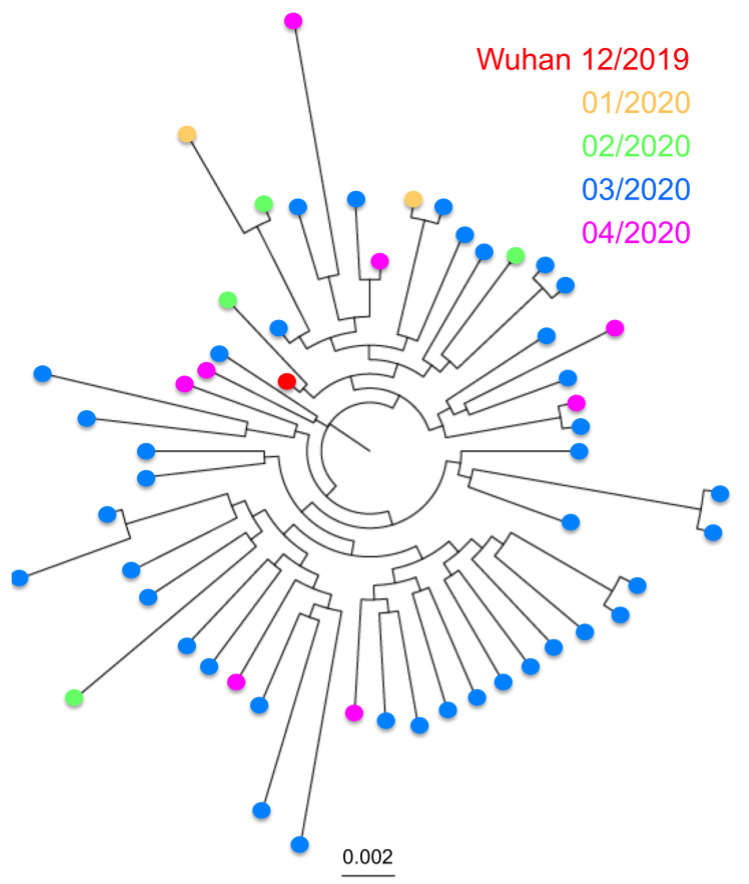
Phylogenetic analysis of RBD sequence variants. Variants are color-coded according to the date of isolation of the sequence.

**Table 1 pathogens-09-00584-t001:** Amino acids of SARS-CoV-2 N protein under diversifying selection pressure.

Reference	Position	Variants				Proportion	%	*p* Value
P	13	L	T	R	S	264/16,353	1.589	0.001
G	34	E	V	L	W	15/16,630	0.09	0.006
D *	144	E	H	Y	N	12/16,642	0.072	0.030
S	180	I	G	C	R	21/16,620	0.126	0.032
R	191	G	C	L	S	14/16,635	0.084	0.005
R	209	K	I	T	del	29/16,616	0.174	0.016
A	381	V	T	P	S	9/16,519	0.054	0.023
Q	28	H	E	R		12/16,637	0.072	0.027
P	151	L	S	H		13/16,633	0.078	0.024
R	185	H	L	C		25/16,623	0.15	0.028
R ^+^	203	K	S	T		3004/13,585	18.108	0.401
A	208	S	G	del		90/16,558	0.541	0.098
S	232	I	R	T		5/16,440	0.03	0.027
D	377	Y	H	G		17/16,505	0.103	0.032
P	20	S	L			9/16,630	0.054	0.005
D	22	G	Y			53/16,588	0.318	0.008
T	24	N	I			32/16,610	0.192	0.076
A	119	S	V			32/16,609	0.192	0.099
S	190	G	I			32/16,608	0.192	0.002
S	202	I	N			80/16,561	0.481	0.066
T	205	I	del			51/16,586	0.307	0.078
A	218	S	V			3/16,635	0.018	0.099
H	300	Q	Y			9/16,469	0.055	0.099
P	344	S	L			20/16,551	0.121	0.040
D	348	H	Y			6/16,607	0.036	0.034
E	378	Q	K			7/16,516	0.042	0.098
A	397	S	V			4/16,513	0.024	0.098

* indicate site(s) included in predicted epitope(s). ^+^ Site presenting no significant selection pressure, but its variants were present at high frequency.

**Table 2 pathogens-09-00584-t002:** Amino acids of SARS-CoV-2 S protein under diversifying selection pressure.

Reference	Position	Variants					Clade	Proportion	%	*p* Value
D	215	N	H	G	S	Y	5	12/17,556	0.068	0.071
Q ^#^	675	K	R	H	S		1, 5	30/17,709	0.169	0.049
V	615	I	L	F			1, 3	16/17,799	0.090	0.068
S ^+^	221	P	L	W			5	16/17,560	0.091	0.251
Q	239	K	R	H			5	16/17,577	0.091	0.056
**V**	**483**	**A**	**F**	**I**			**1, 5**	**33/17,025**	**0.194**	0.017
V	622	I	F	L			3	16/17,792	0.090	0.084
S	943	T	I	P			2	28/17,689	0.158	0.012
A *	1078	S	V	T			1, 4	27/17,782	0.152	0.019
H	49	Y	Q				1, 2	36/17,722	0.203	0.074
**N**	**354**	**K**	**D**				**1**	**5/17,668**	**0.028**	0.037
**H**	**519**	**Q**	**P**				**1, 2**	**3/17,014**	**0.018**	0.079
A	653	S	V				3	3/17,734	0.017	0.076
A ^#^	684	T	V				3	5/17,701	0.028	0.037
A	771	S	V				1, 6	9/17,732	0.051	0.065
A	892	S	V				1	6/17,761	0.034	0.072
D	936	Y	H				2	95/17,750	0.535	0.047
S	940	F	T				2	7/17,747	0.039	0.086
G	1167	S	V				1	5/17,755	0.028	0.060
K	1192	N	Q				1	8/17,729	0.045	0.057
L	5	F					2, 6	102/17,416	0.586	0.020
L	8	V					1	54/17,445	0.309	0.086
A	288	S					4	4/16,067	0.025	0.074
E	309	Q					1, 4	6/15,759	0.038	0.066
**V**	**367**	**F**					**1, 3**	**21/17,577**	**0.119**	0.051
**N**	**439**	**K**					**5**	**40/17,282**	**0.280**	0.024
**G**	**476**	**S**					**1, 5**	**10/17,027**	**0.059**	0.059
**S**	**494**	**P**					**5**	**6/17,023**	**0.035**	0.079
D	614	G					1, 2, 3, 6	11,326/17,744	63.830	0.001
A	706	V					1,3	12/17,695	0.068	0.076
G	1124	V					1, 3	50/17,747	0.282	0.017

**Sites in bold are localized within the RBD.** * indicate site(s) included in predicted epitopes. ^#^ sites included in the furin cleavage region. ^+^ site presenting no significant selection pressure but extensive variation.

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
