# Peer review of "Polymorphism and Selection Pressure of SARS-CoV-2 Vaccine and Diagnostic Antigens: Implications for Immune Evasion and Serologic Diagnostic Performance"

_pathogens, 2020, doi:10.3390/pathogens9070584_

Round 1

Reviewer 1 Report

The paper "Polymorphism and selection pressure of SARS-CoV-2 vaccine and diagnostic antigens: implications for immune evasion and serologic diagnostic performance " investigates polymorphism of the genes coding for the antigen more employed for diagnosis and vaccine development. The paper is well written.

All the parts of the paper are clear and concise. The rewiever only suggest to report if possible the P of the statistical significance when needed (table 1 and 2) and to explain in materials and methods the criteria for identifying the clade, in order to widen the readers audience also to scientists not completely into genetic analysis.

Author Response

Reviewer 1

Comments and Suggestions for Authors

The paper "Polymorphism and selection pressure of SARS-CoV-2 vaccine and diagnostic antigens: implications for immune evasion and serologic diagnostic performance " investigates polymorphism of the genes coding for the antigen more employed for diagnosis and vaccine development. The paper is well written.

All the parts of the paper are clear and concise. The rewiever only suggest to report if possible the P of the statistical significance when needed (table 1 and 2) and to explain in materials and methods the criteria for identifying the clade, in order to widen the readers audience also to scientists not completely into genetic analysis.

ANSWER: We have added P values for each site in Table 1 and 2 as suggested. We also explain in the methods that Clades were identified according to tree structure and branch support measures.

Reviewer 2 Report

Line 28: change “pandemic” to “pandemics”

Line 31: Needs reference

Line 35: Instead of “next few months” you should reference the actual dates, or something like: “available by the end of 2020”

Line 35: Change “would be” to “will be”

Line 45: Combine these two sentences “These are based…”and “A first RNA…”

Line 54: add “in” between “changes” and “SARS-CoV-2”

Line 55: “virus RNA dependent…” to “virus’s RNA-dependent…”

Line 59: drop “on these”

Line 60: give date range, not “until early April 2020”

Line 62: move “here” to the first word

Line 63: We “then” examined the implications…

Line 66: This first paragraph is a little scattered grammatically and doesn’t flow well. I would actually combine the first and second paragraphs

Line 88: Saying “the N protein is slowly diverging” feels too solid. “Appears to be slowly diverging” would be a better wording

Line 90: “Site…” feels like it needs a reference

Line 95: Drop “this is”

Line 108: “appeared less frequent over time”. Also this last sentence, I’m having trouble understanding how/where the data shows this

Line 109: mention which protein sequence first (so say S glycoprotein, not ‘the protein sequence’. Is it “along” the protein sequence, or ‘of’?

Line 150: change to “diagnostics”

Line 192: “that” to “which”

Line 193: lose the comma after strains

Line 193: Move “here” to beginning of sentence

Line 214: give a date range, not “past weeks/months”

Line 226: Careful with blanket statements like “future studies will…”. You need a caveat word in there, like “hopefully” or “potentially”

Line 231: lose “so far”

Line 240: immune selection and human host adaptation are certainly plausible as the selective pressure. You could reference a similar paper that delves into this more

Line 258/259: This last sentence feels weird

Line 312: extra period at the end

Author Response

Reviewer 2

Comments and Suggestions for Authors

Line 28: change “pandemic” to “pandemics”

ANSWER: spelling has been corrected

Line 31: Needs reference

ANSWER: Reference was added.

Line 35: Instead of “next few months” you should reference the actual dates, or something like: “available by the end of 2020”

ANSWER: The wording has been changed as suggested.

Line 35: Change “would be” to “will be”

ANSWER: The wording has been changed as suggested.

Line 45: Combine these two sentences “These are based…”and “A first RNA…”

ANSWER: we prefer to avoid long sentences and have kept these two sentences separate.

Line 54: add “in” between “changes” and “SARS-CoV-2”

ANSWER: The wording has been changed as suggested.

Line 55: “virus RNA dependent…” to “virus’s RNA-dependent…” 

ANSWER: The wording has been changed as suggested.

Line 59: drop “on these”

ANSWER: The wording has been changed as suggested.

Line 60: give date range, not “until early April 2020”

ANSWER: The date range is now provides as suggested.

Line 62: move “here” to the first word

ANSWER: The wording has been changed as suggested.

Line 63: We “then” examined the implications…

ANSWER: The wording has been changed as suggested.

Line 66: This first paragraph is a little scattered grammatically and doesn’t flow well. I would actually combine the first and second paragraphs

ANSWER: The first paragraph has been reworded for better clarity and flow, as suggested, but not combined with the following paragraph.

Line 88: Saying “the N protein is slowly diverging” feels too solid. “Appears to be slowly diverging” would be a better wording

ANSWER: The wording has been changed as suggested.

Line 90: “Site…” feels like it needs a reference

ANSWER: References have been added.

Line 95: Drop “this is”

ANSWER: The wording has been changed as suggested.

Line 108: “appeared less frequent over time”. Also this last sentence, I’m having trouble understanding how/where the data shows this 

ANSWER: Figure 3A shows the different clades color-coded by month of collection, which provides information on changes in sequence abundance over time. The figure is now referenced for greater clarity.

Line 109: mention which protein sequence first (so say S glycoprotein, not ‘the protein sequence’. Is it “along” the protein sequence, or ‘of’?

ANSWER: The wording has been changed as suggested.

Line 150: change to “diagnostics” 

ANSWER: The spelling has been corrected.

Line 192: “that” to “which”

ANSWER: The wording has been changed as suggested.

Line 193: lose the comma after strains

ANSWER: The coma has been deleted.

Line 193: Move “here” to beginning of sentence

ANSWER: The wording has been changed as suggested.

Line 214: give a date range, not “past weeks/months”

ANSWER: We now provide a date range as suggested.

Line 226: Careful with blanket statements like “future studies will…”. You need a caveat word in there, like “hopefully” or “potentially”

ANSWER: The wording has been changed as suggested.

Line 231: lose “so far” 

ANSWER: The wording has been changed as suggested.

Line 240: immune selection and human host adaptation are certainly plausible as the selective pressure. You could reference a similar paper that delves into this more

ANSWER: References have been added, as suggested.

Line 258/259: This last sentence feels weird

ANSWER: The sentence has been corrected.

Line 312: extra period at the end

ANSWER: The period has been deleted.